# Development of a Self-Viscosity and Temperature-Compensated Technique for Highly Stable and Highly Sensitive Bead-Based Diffusometry

**DOI:** 10.3390/bios12060362

**Published:** 2022-05-25

**Authors:** Wei-Long Chen, Han-Sheng Chuang

**Affiliations:** 1Department of Biomedical Engineering, National Cheng Kung University, Tainan 701, Taiwan; alex821214@gmail.com; 2Core Facility Center, National Cheng Kung University, Tainan 701, Taiwan; 3Medical Device Innovation Center, National Cheng Kung University, Tainan 701, Taiwan

**Keywords:** rotational Brownian motion, translational Brownian motion, particle tracking, empirical mode decomposition, bead-based diffusometry

## Abstract

Brownian motion, which is a natural phenomenon, has attracted numerous researchers and received extensive studies over the past decades. The effort contributes to the discovery of optical diffusometry, which is commonly used for micro/nano particle sizing. However, the analysis uncertainty caused by the coupling relationship among particle diameter, temperature, and fluid viscosity usually poses a barrier to precise measurement. Preventing random background noise becomes the key to achieving a high level of accuracy in diffusometry detection. Recently, Janus particles have become known as an ideal tool for resolving the rotational Brownian motion. Followed by our previous study, the rotational Brownian motion and the translational Brownian motion can be separately measured using the Janus particles. Accordingly, a simple self-viscosity and temperature-compensated technique based on the delicate removal of temperature and fluid viscosity variations through particle tracking was first proposed in this study. Consequently, the translational Brownian motion was expressed in terms of particle trajectory, whereas the rotational Brownian motion was expressed in terms of the blinking signal from the Janus particles. The algorithm was verified simulatively and experimentally in temperature (10 °C to 40 °C) and viscosity-controlled (1 mPa·s to 5 mPa·s) fields. In an evaluation of biosensing for a target protein, IFN-γ, the limit of detection of the proposed self-compensated diffusometry reached 0.45 pg/mL, whereas its uncertainties of viscosity and temperature were 96 and 15-fold lower than the pure the rotational Brownian motion counterpart, respectively. The results indicated the low-uncertainty and high-accuracy biosensing capability resulting from the self-viscosity and temperature-compensated technique. This research will provide a potential alternative to future similar bead-based immunosensing, which requires ultra-high stability and sensitivity.

## 1. Introduction

Brownian motion can be performed in two ways, namely, translational and rotational Brownian motion. The former is commonly investigated in a wide variety of applications [1,2,3,4,5,6,7,8,9,10,11,12,13], whereas the latter receives less attention because of its difficulty in realization. However, the rotational Brownian motion is inversely proportional to cubic particle diameter [14], which makes it highly sensitive to particle size change. This ultra-high sensitivity promotes the capability of the rotational Brownian motion to deal with trace target detection [15]. Non-spherical particles are typically used to evaluate the degree of the rotational diffusivity and visualize the rotational Brownian motion. By measuring the temporal angular changes of the non-spherical particles, diffusivity can be quantified [16,17]. However, the monitoring of angular change heavily relies on image quality. Uncertainty escalates when the nominal particle movement cannot be explicitly resolved by a camera or an image recorder. In addition, the physical behavior of non-spherical particles appears to be complicated. Alternatively, colloidal dimers, tetrahedral clusters, diamond nanoparticles, and spherical Janus particles [18,19,20,21] are extensively developed to facilitate the investigation of the rotational Brownian motion. Among them, spherical Janus particles have drawn considerable attention in recent years and have shown potential capabilities because of their simple fabrication, high stability, and low demand for recording hardware. Considering that the rotational Brownian motion based on Janus particles yields signal only associated with temporal intensity variation, the requirement of image quality can be simplified, thereby lowering the uncertainty of angular change. Followed by our previous studies [15,22], the rotational Brownian motion is expressed in terms of blinking signal with spherical Janus particles. The detection of trace target biomolecules and micro-volume viscosity is demonstrated using Janus particles. Nevertheless, background noises, such as viscosity and temperature, often raise unexpected fluctuations during measurement, leading to compromised repeatability and sensitivity.

In addressing the problem, we proposed a method, namely, the self-viscosity and temperature-compensated technique, which combined the translational and the rotational Brownian motions in this research. Based on the Stokes–Einstein equation and Stokes–Einstein–Debye equation [23], their background viscosity and temperature can be eliminated by dividing the translational diffusivity by the rotational diffusivity. The ratio of the translational diffusivity to the rotational diffusivity will be proportional to the squared particle diameter. The proposed technique was eventually verified on the basis of experimental and simulated Brownian motion. Notably, the proposed technique aims to improve the current Brownian motion-based biosensors, such that they can be more robust to prevent disturbances from the background viscosity and temperature during the detection of various targets, such as proteins [8,10], nucleic acids [7,13], or even microorganisms [11].

For an experimental verification, suspending Janus particle motions were measured to estimate their translational and rotational diffusivities. Herein, the Janus particles were made of 1 µm fluorescent core polystyrene (PS) particles with half of the surface coated with 15 nm silver and 5 nm gold. Hence, the half-covered fluorescent particles generated a blinking signal while freely suspending in a medium. Particle images were recorded using a digital camera at 50 Hz for 10 s under a fluorescence microscope to obtain essential Brownian motion. In addition, the algorithm was verified under simulated and experimental conditions in temperatures and viscosities varying from 10 °C to 40 °C and 1 mPa·s to 5 mPa·s, respectively. Both the experimental and simulated data proved that the proposed technique could prevent fluctuations in the measurement of the background (viscosity and temperature). Finally, the self-compensation algorithm was applied to the immunoassay for discussing the dependency of particle size, the translational Brownian motion, and the rotational Brownian motion (Figure 1). The limit of detection (LOD) of the proposed diffusometry reached 0.45 pg/mL. In particular, the uncertainties of the proposed technique of viscosity and temperature were 96- and 15-fold less than that of pure rotational [22] or translational [4,8,9,10] Brownian motion. The proposed algorithm provides a stable and low-noise capability to perform the precise detection of bead-based immunoassays.

## 2. Materials and Methods

### 2.1. Derivation of Self-Compensated Particle Diameter

In this research, a sandwiched bead-based immunoassay was applied to detect a target protein, namely, interferon-gamma (IFN-γ). A total of 500 nm PS-modified particles were conjugated with 1 µm Janus particles to form an immunocomplex in the presence of the target antigen, thereby increasing the particle diameter. Notably, the small PS particles were used to amplify the geometric change. The translational and the rotational diffusivities had to be measured from the same Janus particles to ensure that the calculated particle diameter was free from the background temperature and viscosity fluctuations. The translational diffusivity (*D_t_*) was obtained from the mean particle trajectories, whereas the rotational diffusivity (*D_r_*) was obtained from the time-dependent blinking signal of the Janus particles. Subsequently, the final particle diameter change could be derived from the ratio of the translational diffusivity to the rotational diffusivity (Equation (1), Figure 2):(1)$$\frac{D_{t}}{D_{r}}=\frac{d_{p}^{2}}{3}\;\Rightarrow \;d_{p}=\sqrt{\frac{3D_{t}}{D_{r}}}$$
where *d_p_* indicates the equivalent particle diameter. The translational and the rotational Brownian motions were expressed in terms of diffusivities defined by the Stokes–Einstein equation [14,24] (Equation (2)) and the Stokes–Einstein–Debye equation [14,24] (Equation (3)):(2)$$D_{t}=\frac{k_{B}T}{3\pi \eta d_{p}}=\frac{{\langle x\rangle }^{2}}{4\Delta t},$$
(3)$$D_{r}=\frac{k_{B}T}{\pi \eta d_{p}^{3}},$$
where *k_B_*, *T*, Δ*t*, 〈*x*〉, and η denotes the Boltzmann constant, the temperature, the time interval, a mean trajectory of the particle, and the fluid viscosity, respectively. Both diffusivities were functions of the temperature, viscosity, and particle diameter.

#### 2.1.1. Particle Tracking for Trajectory

The translational diffusivity was obtained by tracking particles for their trajectories. A particle displacement was derived from their coordinates in two different time points multiplied by the pixel size (Figure 2). Subsequently, the particle displacement was used to estimate the translational diffusivity within a predefined time interval (∆t). Five hundred consecutive images were recorded in 10 s with a frame rate of 50 Hz.

In the particle tracking, all images were binarized (black: 0; white: 1) to facilitate the search for particle outlines. The coordinate of each particle was defined in the centroid of the particle image area. Notably, the white areas smaller than 10 pixels and their circularity lower than 0.35 (maximum value of circularity: 1) were excluded to avoid some background noise. The circularity of each particle image was defined as follows:(4)$$circularity=\frac{4\pi \times area\;of\;the\;pattern}{perimeter\;of\;the\;pattern^{2}}$$

The circularity that approaches 1 means the pattern is close to a perfect circle. In addition, overlaps from adjacent particles were excluded from the calculation when their distances between two particles were shorter than 100 pixels. To this end, all particles’ coordinates were located in advance. Subsequently, distances between any two particles were obtained. This action was simply achieved by running the function “pdist” built in MATLAB. Each tracked particle was fixed within a region of interest (ROI) measuring 200 pixels × 200 pixels during tracking (Appendix A).

#### 2.1.2. Time-Dependent Blinking Signal

In addition to the translational Brownian motion, Janus particles produced signals in response to the rotational Brownian motion. The rotational Brownian motion was simply obtained from the blinking behavior of the Janus particles. By tracking particles, their blinking signal was converted into gray-level fluctuations with regard to time. A gray-level intensity at each time point was determined by averaging the gray intensities over the window of 31 pixels × 31 pixels centered at the tracked particle (Appendix A). The measured blinking signal should be further processed to facilitate the extraction of the pure rotational Brownian motion to remove noise resulting from the white noise, electric noise (60 Hz), low-frequency natural background noise, such as in-focus and out-focus intensity trend, and human activities (Appendix A). Notably, the low-frequency noise was not able to be distinguished through Fast Fourier Transform (FFT), especially when the particles were suspended in a high-viscosity solution. A well-designed filter was necessary for this circumstance. In the research, the blinking signal was decomposed into several sub-waveforms, namely, intrinsic mode functions (IMFs), by empirical mode decomposition (EMD) [25,26]. Notably, the EMD is a signal-processing algorithm used to decompose a complex waveform (i.e., multiple frequency modes) to simple components. The decomposed IMFs were expressed in a series of frequencies from high to low [27].

Considering that the blinking frequency reflected a combinative response of the ambient temperature, fluid viscosity, and particle diameter, the embedded IMFs can be altered accordingly. In general, low temperature, high viscosity, or a large particle tended to decrease the rotational Brownian motion, thereby reducing the blinking frequency and vice versa. By comparing the IMFs with the original blinking signal, the IMF showing the highest similarity indicated the optimal waveform that carried the pure blinking signal (Figure 2).

### 2.2. Experimental Setup

The initial concentration of the Janus particles was adjusted to 2 × 10^9^ particles/mL. The particle suspension was first diluted five times to prevent particle aggregation and overlap before use. Five microliters of the diluted particle suspension were pipetted on a glass slide and then topped with a glass cover for subsequent measurement. Brownian motion was measured under a fluorescence microscope (IX73, Olympus, Tokyo, Japan) using a 40 × objective lens (NA: 0.6) and green light (630 nm). Particle images were captured under 50 Hz for 10 s by a digital camera (2048 pixels × 1536 pixels, BFS-U3-31S4C-C, FLIR Blackfly S, Boston, MA, USA). For the viscosity assay, the viscosity of the particle suspension was controlled by adding glycerol [28]. In this study, 25% and 50% (*w*/*w*) of glycerol solutions were used. For the temperature assay, a thermoelectric cooler (TE cooler, TEC1-12706, T-Global, Taoyuan, Taiwan) was used to control the temperature. A droplet of particle suspension was placed on the cooling side to decrease the temperature below room temperature (~25 °C). A cooling fan was placed on the hot side of the TE cooler to prevent the breakdown of the TE cooler under high voltage. On the contrary, a droplet of particle suspension was placed on the heating side of the TE cooler to increase the temperature above room temperature. After acquiring images, consecutive images were analyzed using the abovementioned particle tracking and blinking signal decomposition to derive their translational and rotational diffusivities.

### 2.3. Preparations of Functionalized Janus Particles

#### 2.3.1. Fabrication of Janus Particles

First, 1 μm fluorescent PS particles (F13083, Thermo Fisher, Waltham, MA, USA) were mixed with 95% ethanol, and then spread over a glass slide by drop deposition to form a single layer of particles to obtain the desired Janus particles. The high-purity ethanol promoted rapid evaporation to prevent particle aggregation. After complete drying, the glass slide was transferred to an e-beam evaporator (ULVAC, VT1-10CE, Taipei, Taiwan) for metal deposition. A 15 nm silver film followed by another 5 nm gold film was coated on the PS particles under a coating rate of 1 Å/s [15,29,30]. Next, the glass slide was immersed in a 55 mL centrifugal tube full of deionized (DI) water mixed with 0.1% of Tween-20, and then the coated particles were collected from the glass slide by sonication. The collected particles were purified with filter disks (pore size: 3.2 µm) five times to remove aggregates and metal debris. The final concentration of the purified particle suspension was adjusted to 2 × 10^9^ particles/mL in DI water [15].

#### 2.3.2. Sandwiched Immunocomplexed Janus Particles

The gold coating of the Janus particles partially blocked the fluorescence and enabled antibody conjugation via the thiol bond. Accordingly, the monoclonal anti-IFN-γ IgG (orb26592, Biorbyt, Cambridge, UK) could be conjugated with the Janus particles through a gold conjugation kit (ab154873, Abcam, Cambridge, UK). The conjugate reaction took place in a 10 mM amine-free buffer, with a pH range between 6.5 and 8.5 for all antibodies. The antibodies were diluted to 0.1 mg/mL of the diluent buffer in the kit. Twelve microliters of the diluent antibodies were mixed with 42 µL of the gold reaction buffer. Subsequently, the mixture was incubated with the Janus particles in a shaker (400 rpm) for 20 min at room temperature (~25 °C). Five microliters of the gold quencher buffer were mixed gently after incubation. The antibodies were covalently attached on the surface of the Janus particles through lysine residues.

In amplifying particle geometric changes, 500 nm PS particles (07763-5, Polysciences, Niles, IL, USA) were further used to form a sandwiched immunocomplex in the presence of target antigens. Consequently, monoclonal anti-IFN-γ IgG was conjugated with the amine-modified 500 nm PS particles. Then, 10 μL of the particle suspension was washed with 200 µL of 50 mM 2-(N-morpholino)ethanesulfonic acid buffer (MES buffer) at pH 5.5. Meanwhile, 10 µL of 0.5 mg/mL antibodies was activated by mixing 2 µL of 10 mg/mL 1-ethyl-3-(3-dimethylaminopropyl)-carbodiimide, 4 µL of 10 mg/mL N-hydroxysuccinimide, and 30 µL of MES buffer in a shaker for 15 min at room temperature. Next, the mixture was incubated with particle suspension in a shaker (800 rpm) for 4 h at 4 °C. After storing at 4 °C overnight, the suspension was washed with PBS with 1% Tween-20 (PBST) three times and then blocked with 1% BSA for 1 h.

After preparing the functionalized Janus particles and the modified 500 nm PS particles, 20 µL of the functionalized Janus particle suspension was incubated with 10 mL of different concentrations (1 pg/mL, 100 pg/mL, 10 ng/mL, and 1 µg/mL) of the IFN-γ antigen (orb82062, Biorbyt, Cambridge, UK) in a shaker for 1 h at room temperature. Subsequently, the mixture was washed thrice with PBST to remove unbound antigens, followed by incubating with 10 µL of the modified PS particle suspension for 1 h at room temperature. Afterward, the Janus particles were conjugated with the modified PS particles to form a sandwiched immunocomplex. The sandwiched immunocomplexed particle diameter basically grew with the increased antigen concentration (Appendix A). In addition, the target antigen could be changed by replacing the conjugated antibodies on both the Janus particles and the PS particles.

### 2.4. Generation of Simulated Images

Simulated particle images were used in this study to investigate the effects under ideal conditions and verify the self-viscosity and temperature-compensated technique. For the generation of simulated particle images, some parameters, including the pixel size and the frame rate of the camera, the magnification and the numerical aperture of the objective lens, the fluid temperature and viscosity, the excitation wavelength, and the particle diameter, were considered (Table 1) [31]. Among the abovementioned parameters, the fluid temperature and viscosity were coupled with each other, and their relationship was approximated by a four-parameter exponential [32] (Appendix A). A single particle image was derived from a 2D Gaussian profile with a maximum intensity of 255 [33]. Notably, the window size for each particle must be sufficient to include the entire diffusive trajectory of the particle during measurement. The movement in each frame was obtained using Equation (5) [34]:(5)$$\langle x^{2}\rangle =2D_{t}t,$$
where *t* indicates the time according to the frame rate. The blinking signal was first generated on the basis of a sinusoidal wave to simulate the rotational Brownian motion, which contained the information of temperature, fluid viscosity, and particle diameter. Next, the sinusoidal wave was coupled with two random background fluctuations and low-frequency and high-frequency noises to mimic the real measurement [35]. The low-frequency noise indicated white noise, whereas the high-frequency noise indicated electronic noise. The synthetic intensity waveform was used to simulate the blinking images of the Janus particles (Appendix A). Particles within an ROI window of 200 pixels × 200 pixels were randomly moving.

## 3. Results and Discussion

### 3.1. Determination of Particle Diameter out of Rotational Diffusivity and Translational Diffusivity

The simulated images generated under fluid viscosity of 1 mPa·s and particle diameter of 1 μm at 25 °C without background noise were investigated first. EMD was applied to decompose the original signal into four IMFs (Figure 3a). A 1D cross-correlation algorithm was applied to analyze the similarity between the IMFs and the original signal to identify the optimal waveform that can express the major traits out of the original signal. The similarity values were automatically calculated and compared on MATLAB. In principle, a low-ranked IMF could show a high similarity to a high-ranked IMF in the absence of noise. Moreover, in the absence of noise during the simulation, IMF 1 retained the major traits out of the original blinking signal rather than the other IMF modes. IMF 1 reached the highest similarity, up to 0.98 (Figure 3b). The similarity value indicated the correlation between the IMFs and the original signal, where 1 indicated the highest similarity and vice versa. However, in the presence of noise, the optimal IMF was shifted to a higher rank because the high-frequency noise will occupy the low-ranked IMF modes (Figure 3c). Consequently, IMF 3 could resemble the original blinking signal compared with the other IMF modes (Figure 3d). In addition, waveforms may become ambivalent when the medium has high viscosity (5 mPa·s and 25 °C, Figure 3e). The noise was observed in all low-ranked IMF modes (IMF 1-IMF 3). Considering that the blinking signal was dampened by viscosity, the optimal IMF was shifted to a higher-ranked mode (IMF 4, Figure 3f). Based on the abovementioned results, the high-frequency components of the original signal contributed to the low-ranked IMFs, whereas the low-frequency components contributed to the high-ranked IMFs. In the case of low viscosity or high temperature, the low-ranked IMF can account for the original signal. Conversely, the high-ranked IMF retained most traits of the original signal under a high-viscosity or a low-temperature condition.

After examining the similarity of each IMF based on the 1D cross-correlation algorithm, the optimal IMFs with the highest similarity to the original signal in all simulated conditions derived their blinking frequencies in response to the rotational diffusivities by Fast Fourier Transform (FFT). The FFT spectrum was fitted with a two-term Gaussian model to determine the peak frequency (Figure 4a–c) [36]. The blinking frequency under a fluid viscosity of 1 mPa·s and particle diameter of 1 µm at 25 °C with background noise was approximately 1.38 Hz, which was close to the theoretical value of 1.43 Hz (Figure 4a). However, the blinking frequency dropped to 0.26 Hz when the viscosity was increased to 5 mPa·s (Figure 4b). In the case that the temperature increased to 40 °C, the blinking frequency escalated to 2.13 Hz (Figure 4c).

The particle trajectory was also required based on Equation (1) to estimate the final particle diameter. Consequently, particle displacements were simulated under the same conditions (Figure 4d). The displacements were simulated based on Equation (5) by adjusting *D_t_* [37]. The red line indicates a particle trajectory plotted under 1 mPa·s at 25 °C. At a high temperature (40 °C), the particle trajectory (black line) fluctuated. However, the particle trajectory (blue line) became mild when a relatively high viscosity (5 mPa·s) was involved.

### 3.2. Evaluations of Simulated Bead-Based Brownian Motion with Viscosity and Temperature Changes

For evaluations, the simulated images under viscosity and temperature changes were investigated using the proposed technique, pure translational diffusometry, and pure rotational diffusometry. More than 100 particles were individually tracked for their rotational and translational diffusivities to obtain the diameter of the particles. The particle diameter distribution was plotted on the basis of the probability density function using the Kernel density estimate (KDE) [38]. Relative variation derived from the normalization of the first group (without noise) was used to discuss the changes in different conditions. The noise slightly altered the particle diameter distribution and the standard deviation increased in the three algorithms (Figure 5a,b). However, with regard to the viscosity and temperature variations, *D_r_* and *D_t_* declined more than 45% when the viscosity was two times as high as the control group or the temperature change exceeded a 30 °C increase (Figure 5c–f). Notably, the coupling effect between the temperature and viscosity should be carefully considered in the simulation. Therefore, the viscosity must be adjusted in response to the temperature change. For water solutions, the relationship between the viscosity and temperature was approximated herein by a four-parameter exponential [38]. By contrast, the particle diameter derived from the proposed technique was more stable and reliable, because the maximum variations in temperature and viscosity changes were no more than 3.31%. In addition, different particle diameters were investigated simulatively. Moreover, *D_r_* declined 3.65-fold and 8.78-fold when the particle diameter became 1.50-fold and 2.00-fold larger than the control group (d_p_ = 1 µm), respectively (Figure 5g,h). These values were in good agreement with the theoretical predictions, which were 3.38 and 8.00, respectively. The results indicated that the proposed technique can be used to derive the particle diameter free from the influences of the background viscosity and temperature variations.

### 3.3. Evaluations of Experimental Viscosity and Temperature Changes

Although the proposed technique was well confirmed with the simulated images under preset conditions, actual environments might be more complex than the simulation. For example, the local temperature and viscosity of the surrounding particles can be seriously altered by green light due to the surface plasmon resonance [15,39,40]. Considering the uncertainty, the algorithm was also investigated experimentally to evaluate its efficacy. In the study of viscosity, 0%, 25%, and 50% of glycerol in water were prepared to achieve 1, 2, and 6 mPa·s, respectively [28]. The Janus particles were homogeneously resuspended in glycerol solutions to monitor their Brownian motion during the measurement. Subsequently, consecutive particle images were taken under 50 Hz and a 40 × objective lens for 10 s.

After taking the particle images, the EMD of the blinking signal, trajectory, similarity of IMFs, and FFT of IMF were analyzed (Figure 6). The first row of Figure 6a shows the original blinking signal and four IMFs of particles in pure water. IMF 3 was used to estimate the blinking frequency, because its similarity (0.6) was higher than other IMFs (Figure 6b). In this case, *D_r_*, *D_t_*, and the particle diameter were 0.9 Hz, 2.96 × 10^−13^ m^2^/s, and 993.97 nm, respectively (Figure 6c). However, some of these values, except for the particle diameter, were not in good agreement with the theoretical predictions, which were 1.43 Hz, 4.78 × 10^−13^ m^2^/s, and 1000 nm. The significant deviations may be attributed to the boundary effect in the fluid [41]; that is, the viscosity increased when particles were suspended near the boundary. Despite the uncertainty resulting from the boundary effect, the particle diameter derived from the proposed technique remained unaffected, indicating high stability.

**Figure 5 biosensors-12-00362-f005:**
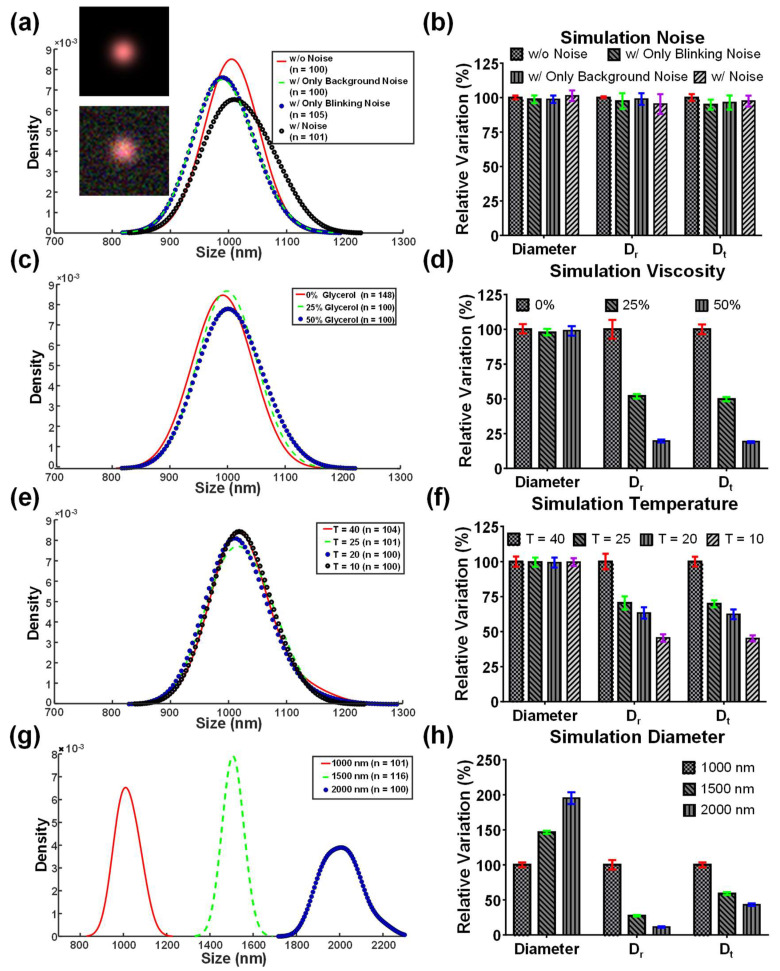
Simulations of the KDE distribution of the particle diameter and relative variation of particle diameter, *D_r_*, and *D_t_*. (**a**,**b**) The KDE distribution and relative particle diameter, *D_r_*, and *D_t_* under different noise. Neither the background noise nor blinking noise affects the particle diameter, *D_r_*, and *D_t_*. The inserted figures are the simulated images without and with the white noise. (**c**,**d**) The KDE distribution and relative particle diameter, *D_r_*, and *D_t_* under different viscosity. *D_r_* and *D_t_* decreased with the increase of viscosity. The viscosity was inversely proportional to *D_r_* and *D_t_*. (**e**,**f**) The KDE distribution and relative particle diameter, *D_r_*, and *D_t_* under different temperatures. *D_r_* and *D_t_* decreased with the decrease of temperature. The temperature was proportional to *D_r_* and *D_t_*. (**g**,**h**) The KDE distribution and relative percentage of particle diameter, *D_r_*, and *D_t_* of different particle diameters. *D_r_* drops more significantly than *D_t_* when the particle diameter increases.

When particles were suspended in a 50% glycerol solution, *D_r_* and *D_t_* remarkably decreased because of damping from high viscosity. The signal waveform was shifted to a high-ranked IMF in a relatively high-viscosity fluid (Figure 6d). In this study, IMF 5 was used to stand for the blinking frequency (0.15 Hz) because of its high similarity (0.87) among all the IMF modes (Figure 6e,f). For the investigation of temperature variation, a TE cooler (TEC1-12706, T-Global) was used to create a temperature field in particle suspension. When the particle suspension was set at 40 °C, the blinking signal fluctuated more than that at 25 °C (Figure 6g). In addition, the optimal IMF shifted to a low rank (IMF 2, Figure 6h). A blinking frequency of 1.25 Hz was eventually estimated for the temperature field at 40 °C (Figure 6i). The particle trajectories under each condition are shown in Figure 6j. The red, blue, and black lines represent particle trajectories measured in pure water at 25 °C, in 50% glycerol solution at 25 °C, and in pure water at 40 °C, respectively. Evidently, the particle trajectory fluctuated more at high temperatures, but turned mild in a high-viscosity fluid.

The maximum relative variations of *D_r_* and *D_t_* in 25% and 50% glycerol solutions were 52% and 21%, respectively (Figure 7a). These values were close to the theoretical value of 50% and 16%, which referred to 2 and 6 mPa·s, respectively. However, the particle diameters were estimated to be 1034.96 ± 68.02 nm and 1036.63 ± 64.21 nm in the two abovementioned glycerol solutions (Figure 7b). Unlike the pure translational diffusivity and the pure rotational diffusivity, the particle diameter expressed relatively high stability and showed a maximum variation of only 7.9%. When the temperature fields were at 40 °C, 25 °C, 20 °C, and 10 °C, the particle diameters were estimated to be 1024.74 ± 36.08, 1034.90 ± 54.18, 1032.84 ± 42.07, and 1030.17 ± 38.94 nm, respectively (Figure 7c,d). The experimental results of the viscosity and temperature changes showed good agreement with their simulated counterparts. Notably, the standard deviations of the measured particle diameters under different conditions were all smaller than those of *D_r_* and *D_t_*. The improvement indicated that the noise can be effectively removed by the proposed self-viscosity and temperature-compensated technique.

### 3.4. Realization of Highly Stable and Sensitive Immunosensing

Our previous work indicated that the diameter of immunocomplexed particles increased with the increased antigen concentration based on the sandwiched configuration [15]. The same work was conducted here to demonstrate improvements in stability and sensitivity using the proposed technique. Consequently, the 1 µm Janus particles were conjugated with the anti-IFN-γ IgG and incubated with different concentrations of the IFN-γ (1 pg/mL, 100 pg/mL, 10 ng/mL, and 1 µg/mL) [42]. Afterward, the 500 nm functionalized PS particles were conjugated to form sandwiched immunocomplexes. The self-compensated diffusometry was performed here to estimate the particle diameter. The experimental result showed that the particle diameter depicted a clear and distinct increasing trend with the concentration of antigen as compared with *D_r_* and *D_t_* (Figure 8a). The LOD of the proposed technique that reached 0.45 pg/mL was derived from Figure 8a based on the three-sigma rule. Meanwhile, *D_r_* showed a faster change than *D_t_*, indicating that the rotational Brownian motion was more sensitive than the translational Brownian motion. The LODs of the pure rotational Brownian motion and the pure translation Brownian motion were 9.57 and 384.98 pg/mL, respectively. In addition, the standard deviation of the particle diameter in the proposed technique was increased with the concentration of the antigen, which may be referred to the KDE distribution of the particle diameter (Figure 8b). Compared with the pure rotational Brownian motion, which required recalibration every time when the particles were resuspended in a different solution or at a different temperature, the proposed technique can effectively mitigate the influences from viscosity and temperature. When 1 pg/mL of IFN-γ was experimentally tested under different viscosities and temperatures, the maximum deviations caused by the viscosity and temperature changes in the proposed technique were 96- and 15-fold lower than those in the pure rotational Brownian motion (Figure 8c,d). This result indicates that the proposed diffusometry may serve as a highly stable and sensitive tool for the early diagnosis of diseases.

**Figure 6 biosensors-12-00362-f006:**
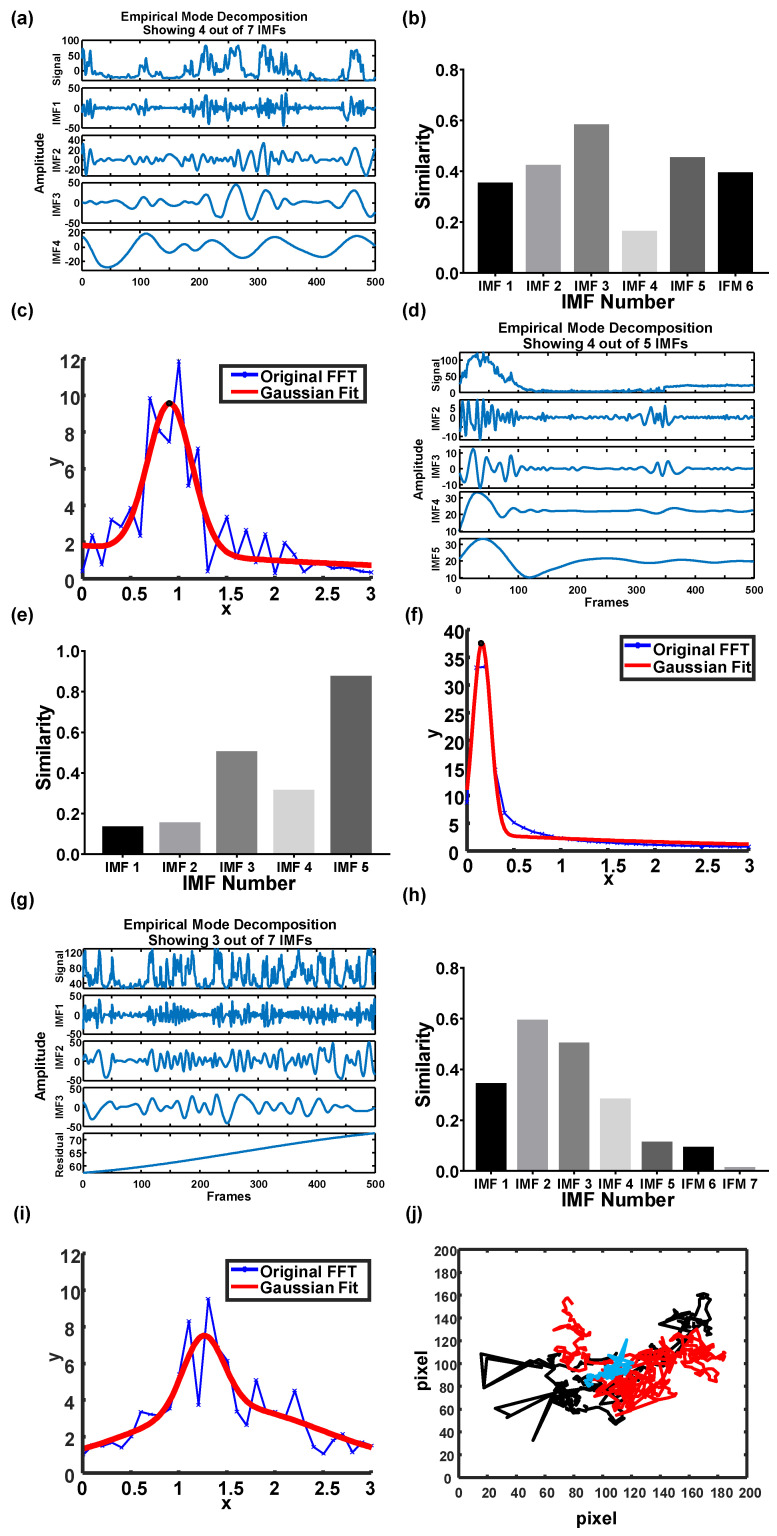
EMD, similarity of IMFs, and blinking frequency of particles under different conditions. (**a**–**c**) The EMD, similarity of IMFs, and blinking frequency of particles under 0% glycerol at 25 °C. (**d**–**f**) The EMD, similarities of IMFs, and blinking frequency of particles under 50% glycerol at 25 °C. (**g**–**i**) The EMD, similarity of IMFs, and blinking frequency of particles under pure water and 40 °C. (**j**) The trajectory of the particles under different conditions. The red, blue, and black lines denote the particles under pure water at 25 °C, 50% glycerol solution at 25 °C, and pure water at 40 °C, respectively.

**Figure 7 biosensors-12-00362-f007:**
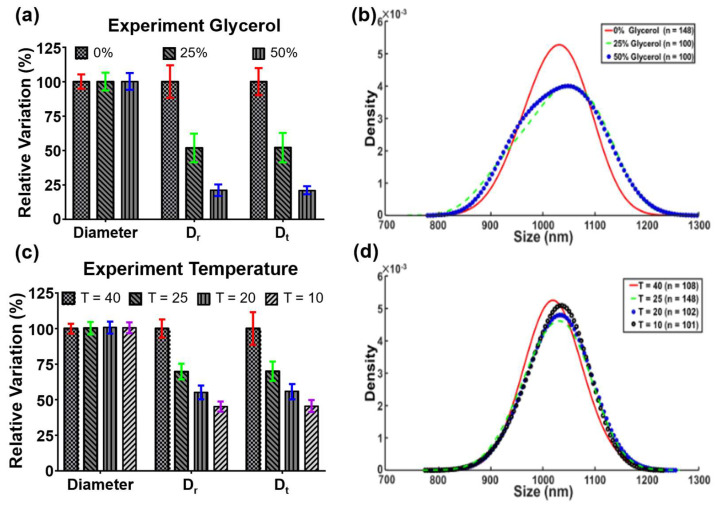
Experimental results. (**a**,**b**) The relative percentage of particle diameter, *D_r_*, and *D_t_*, and KDE distribution of particles under different concentrations of glycerol. *D_r_* and *D_t_* decreased with the increase of glycerol concentration. The viscosity was inversely proportional to *D_r_* and *D_t_*. (**c**,**d**) The relative particle diameter, *D_r_*, and *D_t_*, and KDE distribution under different temperatures. *D_r_* and *D_t_* decreased with the decrease of temperature. The temperature was proportional to *D_r_* and *D_t_*.

**Figure 8 biosensors-12-00362-f008:**
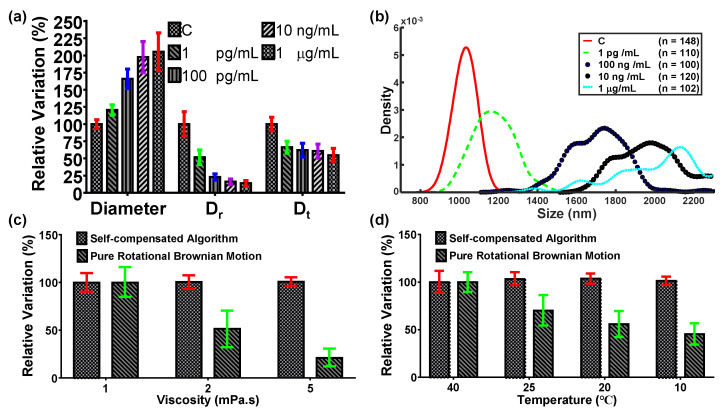
Relative diffusometry and KDE of immunoassay. (**a**,**b**) The relative particle diameter, *D_r_*, and *D_t_*, and particle diameter distribution of different concentrations of IFN-γ. (**c**) Comparison of different viscosities with pure rotational Brownian motion. (**d**) Comparison of different temperatures with pure rotational Brownian motion.

## 4. Conclusions

Brownian motion has been proven as a potential technique for biosensing and has been extensively developed for sensitive detections of trace biomolecules, such as bacteria, proteins, and nucleic acids, in the early phase of diseases. However, the background noises resulting from viscosity and temperature are common disturbing factors, which can pose challenges in achieving highly stable and highly sensitive measurements. Based on the self-viscosity and temperature-compensated technique, the rotational Brownian motion and the translational Brownian motion were simultaneously measured by tracking the 1 µm Janus particles. Subsequently, the unfavorable background noises were removed by dividing the rotational diffusivity by the translational diffusivity. For the translational Brownian motion, the coordinates of each single particle were determined by tracking the particle trajectory. These coordinates were used to lock in the specific ROI to record the blinking signal. For the rotational Brownian motion, the blinking signal was decomposed into a few IMFs by EMD. The final blinking frequency was determined from the major IMF, which yielded maximum similarity. Based on the translational diffusivity and the rotational diffusivity, the final particle diameter was derived on the basis of the self-viscosity and temperature-compensated technique. The viscosity and temperature changes were evaluated simulatively and experimentally in this study. Unlike the pure translational diffusivity or rotational diffusivity, which were seriously altered by the environmental viscosity and temperature, the particle diameter showed not only good agreement with the prediction, but also good stability in all conditions. Thus, a cytokine, namely, IFN-γ, was used to investigate the particle diameter changes with regard to the concentrations ranging from 1 pg/mL to 1 µg/mL. A more linear increase was obtained with the increased concentration compared with the rotational diffusometry. Disturbances caused by the temperature and viscosity in the proposed technique were significantly smaller than other unprocessed counterparts (the pure translational Brownian motion and the pure rotational Brownian motion). An LOD of nearly 0.45 pg/mL was reached. In addition, the maximum deviations in the viscosity and temperature fields were mitigated 96- and 15-fold. This innovative technique successfully improved the existing diffusometry to achieve highly stable and highly sensitive detection of diseases for promising future point-of-care testing.

## Figures and Tables

**Figure 1 biosensors-12-00362-f001:**
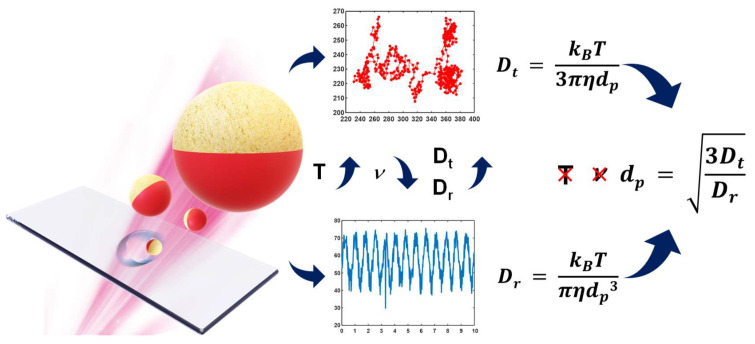
Schematic of self−viscosity and temperature−compensated technique. The translational diffusivity was determined by particle trajectory, whereas the rotational diffusivity was defined by the blinking signal of the Janus particles. The particle size could be estimated from the ratio of the translational diffusivity to the rotational diffusivity. An immunoassay was applied to increase the particle size and verify the proposed technique, because the translational diffusivity and the rotational diffusivity are size-dependent.

**Figure 2 biosensors-12-00362-f002:**
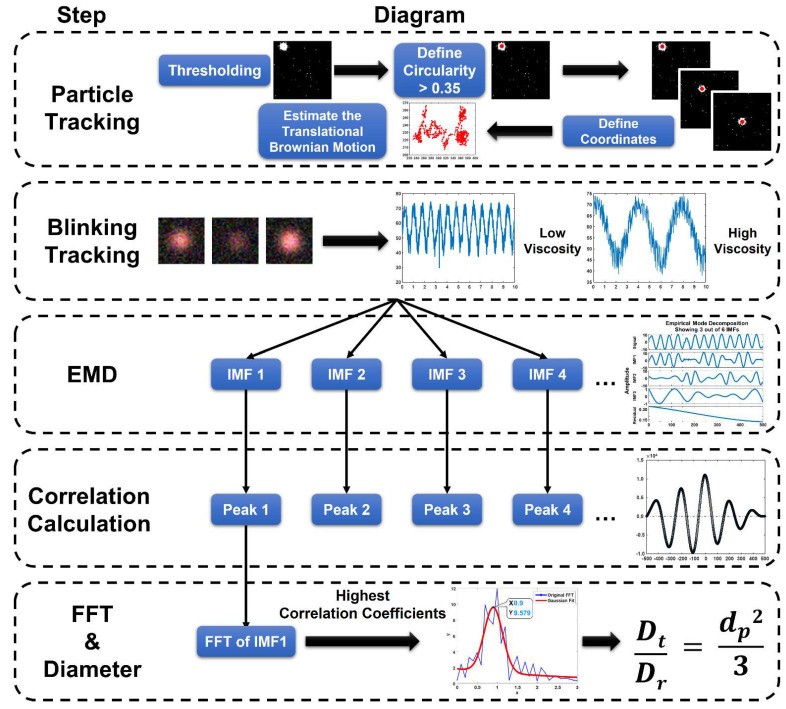
The schematic of the self−viscosity and temperature−compensated technique step. The particle tracking was used to evaluate the translational and the rotational diffusivities. The particle images were binarized by thresholding to define the circularity of each particle. The coordinates of each particle were defined after removing the close particles. The blinking signal of the particle was captured according to particle coordinates. To eliminate the noise of the blinking signal, EMD was applied to decompose the blinking signal into several IMFs. The particle diameter was derived by dividing the translational diffusivity by the rotational diffusivity.

**Figure 3 biosensors-12-00362-f003:**
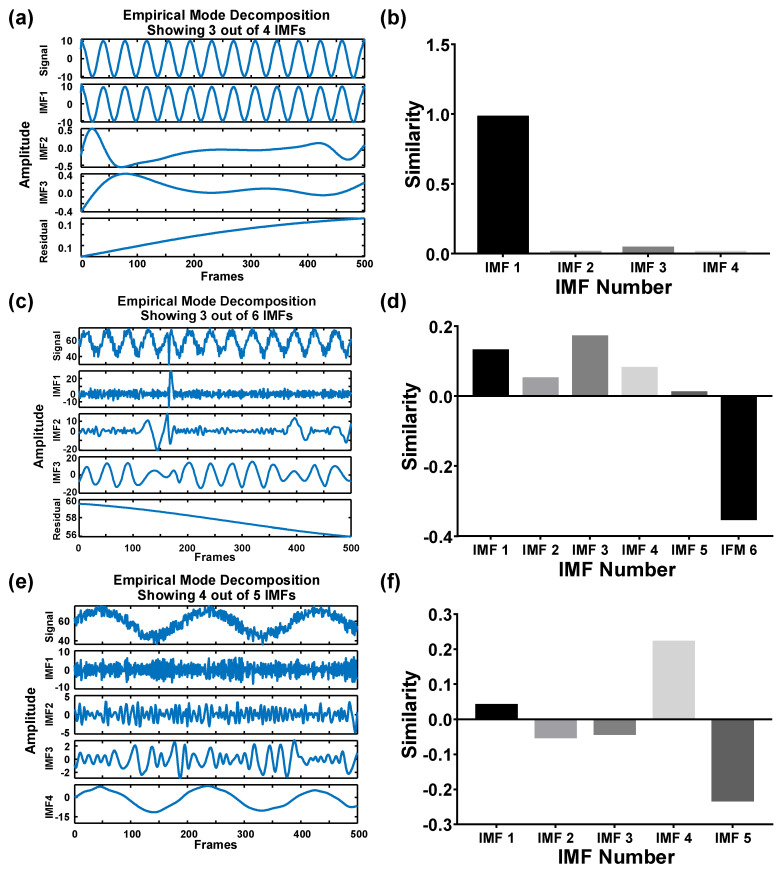
EMD decomposes the noise−blinking signal. (**a**) The signal without noise under 1 mPa·s and 25 °C. Three of the four IMFs are shown. (**b**) The similarity of each IMF to the original signal without noise (**c**) The signal with noise under 1 mPa·s and 25 °C. Three of the six IMFs are shown. (**d**) The similarity of each IMF to the original signal with noise. (**e**) The signal with noise under 5 mPa·s of viscosity. Four of the five IMFs are shown. (**f**) The similarity of each IMF to the original signal w/noise in the 5 mPa·s group.

**Figure 4 biosensors-12-00362-f004:**
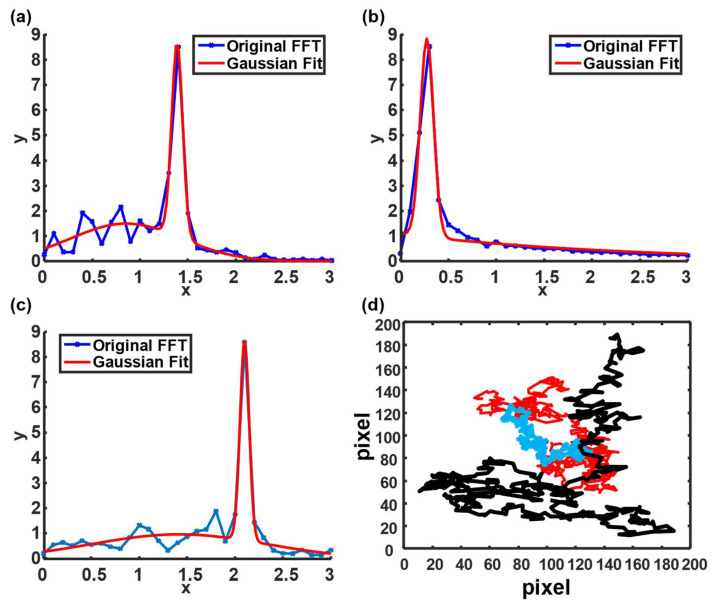
Blinking frequency and particle trajectory under different conditions. (**a**–**c**) Blinking frequency under the following conditions: 25 °C and 1 mPa·s, 25 °C and 5 mPa·s, and 40 °C and 1 mPa·s. (**d**) The particle trajectory of different conditions. Red, blue, and black lines denote the particle trajectories simulated under the following conditions: 25 °C and 1 mPa·s, 25 °C and 5 mPa·s, and 40 °C and 1 mPa·s, respectively.

**Table 1 biosensors-12-00362-t001:** Default parameters for the simulated particle images.

Parameter	Value	Parameter	Value	Parameter	Value
Pixel Size	3.45 µm	Magnification	40×	Temperature	25 °C
Frame Rate	50 Hz	NA ^†^	0.6	Viscosity	1 mPa·s
Ex *	630 nm	Elapsed Time	10 s	dp ^‡^	1 μm

* Excitation wavelength; ^†^ numerical aperture; ^‡^ particle diameter.

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
