# Peer review of "Development of a Self-Viscosity and Temperature-Compensated Technique for Highly Stable and Highly Sensitive Bead-Based Diffusometry"

_biosensors, 2022, doi:10.3390/bios12060362_

Round 1
Reviewer 1 Report
Development of a self-compensated algorithm for highly stable 2and highly sensitive bead-based diffusometry
In this article, authors develop a methodology to detect the presence of a molecule type (interferon gamma) in a medium from diffusometry measurement. To do this, authors made Janus nanoparticles for which the rotational diffusion is observable through fluctuations of the particle intensity. By equating the translational diffusion coefficient with the rational diffusion coefficient, they manage to obtain a measure of mass of individual molecules that is independent of viscosity and temperature. By functionalizing these Janus particles with an antigen, they can target a specific molecule that will bind the particle thus changing the mass. The manuscript is interesting, the concept seems to work, and could be of interest to the readers of 'Biosensors'. I would recommend publication after adding/discussing some points:
- How many IF-G are needed to be bound on the Janus particle start to observe a change in mass using this technique?
- It is claimed that 0.45pg/mL is the lower limit of concentration that can be detected, however these results are not presented. The question is even more valid as nanoparticles are 1micrometer size while interferon gamma is at least 100 times smaller.
- I am not sure to understand in which system this methodology would prove more useful that just looking at western blot for detection of molecules? It might be interesting to mention which system would benefit from such methodology.
- There are problems with most numbers and units appearing in the text, please check everything again and the pdf output as well.
- I think that figure S1 from supplementary is a nice summary of the method and should appear in the main text where it will help the reader follow authors’ ideas.
Line 92: How difficult is it to functionalize other antigens? A brief comment on that would be useful to reader.
Line 122: Can you define ‘circularity’?
Line 124: ‘overlaps from adjacent particles were excluded from the calculation when their distances between two particles were shorter than 100 pixels’.
I am not sure to understand, how were they excluded?
Line 222: ‘and their relationship was approximated by a four-parameter exponential’
Is this approximation an empirical fitting or analytically derived?
Line 227: the equation is not correct. $\langle x \rangle$ stands for the average of x that is 0. The quantity that is calculated is the standard deviation of displacement $\sqrt{\langle x^2\rangle}$.
Line 253: Is the detection of the best IMF manual or automatic?
Video S2: does not work on my VLC
Minor English :
“and applied in a wide variety of applications”-> Repetition of ‘apply’
“Consequently, the rotational and translational diffusivities were delicately correlated to remove the unfavorable background noises.” -> Correlated does not work here.
Author Response
Editorial Board
Biosensors
May. 21, 2022
Dear Editor,
On behalf of the authors, I would like to submit our revised manuscript entitled “Development of a self viscosity and temperature compensated technique for highly stable and highly sensitive bead-based diffusometry” (ID: biosensors-1709634) in response to the reviewers’ comments. As requested by the reviewers, we have clarified all the minor concerns. Our point-by-point responses to the reviewers are also attached below and all the changes are highlighted throughout the annotated manuscript. We hope, with the improvement, you will now find our manuscript suitable for publication in Biosensors. Finally, we thank you and the reviewers for your precious time and advice. Please feel free to contact us anytime with any questions.
Sincerely,
Corresponding Author Information:
Han-Sheng Chuang, Professor
Tel:+886 6-2757575#63433, Fax:+886 6-2343270, oswaldchuang@mail.ncku.edu.tw
Department of Biomedical Engineering, National Cheng Kung University, Tainan, Taiwan
Point by point responses to reviewers’ comments
Reviewer #1
General Comment:
In this article, authors develop a methodology to detect the presence of a molecule type (interferon gamma) in a medium from diffusometry measurement. To do this, authors made Janus nanoparticles for which the rotational diffusion is observable through fluctuations of the particle intensity. By equating the translational diffusion coefficient with the rational diffusion coefficient, they manage to obtain a measure of mass of individual molecules that is independent of viscosity and temperature. By functionalizing these Janus particles with an antigen, they can target a specific molecule that will bind the particle thus changing the mass. The manuscript is interesting, the concept seems to work, and could be of interest to the readers of 'Biosensors'. I would recommend publication after adding/discussing some points:
Response:
We truly appreciate the reviewer’s positive feedback. Regarding some unclear aspects of the manuscript, we have revised them carefully in response to the reviewer’s comments point by point. The detailed responses are addressed as follows.
Comment 1:
How many IF-G are needed to be bound on the Janus particle start to observe a change in mass using this technique?
Response 1:
In general, only size matters in the Brownian motion. To estimate the appropriate concentrations of small particles as well as IFN-γ to be used, the total gold surface area of a large Janus particle was firstly calculated (1.57x10-12 m2). Assume the entire gold surface was fully covered by antibodies (12 nm x 12 nm), the maximum number of 500-nm PS particles attached to the Janus particle was around 6, considering the projected area of a single PS particle was 2.5x10-13 m2. With the assumption, 10 µL of 0.5 mg/mL capture antibody was required to mix with 20 µL of Janus particles (4×107 #/mL).In addition, the equivalent diameter of one 500 nm PS particle conjugated with a 1-μm Janus particle was estimated to be 1.04 μm. However, the standard deviations of pure 1-μm Janus particles were experimentally determined to be 62 nm. For a reliable measurement, at least two IFN-γ proteins would be needed to capture two 500-nm PS particles on a single Janus particle, forming an equivalent diameter of 1.08 μm. The additional information was added in the supporting information as well.
“To estimate the appropriate concentrations of small particles as well as IFN-γ to be used, the total gold surface area of a large Janus particle was firstly calculated (1.57x10-12 m2). Assume the entire gold surface was fully covered by antibodies (12 nm x 12 nm), the maximum number of 500-nm PS particles attached to the Janus particle was around 6 considering the projected area of a single PS particle was 2.5x10-13 m2. With the assumption, 10 µL of 0.5 mg/mL capture antibody was required to mix with 20 µL of Janus particles (4×107 #/mL). In addition, the equivalent diameter of one 500 nm PS particle conjugated with a 1-μm Janus particle was estimated to be 1.04 μm. However, the standard deviations of pure 1-μm Janus particles were experimentally determined to be 62 nm. For a reliable measurement, at least two IFN-γ proteins would be needed to capture two 500-nm PS particles on a single Janus particle, forming an equivalent diameter of 1.08 μm.” (Supporting Information 2)
Comment 2:
It is claimed that 0.45pg/mL is the lower limit of concentration that can be detected, however these results are not presented. The question is even more valid as nanoparticles are 1micrometer size while interferon gamma is at least 100 times smaller.
Response 2
The LOD was actually derived from Figure 8a based on the three-sigma rule. Although the IFN-γ protein is at least 100 times smaller than the 1-µm Janus particle, the final immunocomplex was in sandwich configuration, which is composed of antibody conjugated 1-µm Janus particle, IFN-γ, and antibody-conjugated 500-nm PS particle. Notably, the 500-nm PS particle was used herein to enhance the size change. Therefore, the tiny change of concentration could still be measured from their Brownian motion. An additional statement was added in the manuscript to emphasize this determination of the LOD.
“The LOD of the proposed technique that reached 0.45 pg/mL was derived from Figure 8a based on the three-sigma rule.” (Page 14, Line 444-446)
Comment 3:
I am not sure to understand in which system this methodology would prove more useful that just looking at western blot for detection of molecules? It might be interesting to mention which system would benefit from such methodology.
Response 3:
We appreciate the reviewer for raising this good question. For the detection of proteins, indeed western blotting is a commonly-performed method to see the result. However, it usually suffers from poorly quantitative sensitivity and accuracy. The operation is also time-consuming (~ 0.5-2 h) and highly dependent on some expensive facilities for gel electrophoresis and imaging. By contrast, the diffusometry improves the above drawbacks. Furthermore, the proposed self-compensated diffusometry can even achieve noise-free detection. It should be noted that the technique aims to apply in diffusometry. According to the past literature, 1-7 the diffusometry-based biosensors have been widely used in the detection of various trace proteins, DNAs, or microorganisms, which were highlighted by their low LODs outperforming other counterparts, such as ELISA or western blot. An additional statement was added in the introduction to clarify the use of this technique.
“Notably, the proposed technique aims to improve the current Brownian motion-based biosensors, such that they can be more robust to prevent disturbances from the background viscosity and temperature during the detection of various targets, such as proteins [24,25], nucleic acids [26,27], or even microorganisms [28].” (Page 2, Line 68-71)
Comment 4:
There are problems with most numbers and units appearing in the text, please check everything again and the pdf output as well.
Response 4:
We apologize for the incorrect units and symbols in the text. The incorrect symbols and units, such as γ, µ, Stokes, were corrected
“Keywords: Rotational Brownian motion; translational Brownian motion; particle tracking; empirical mode decomposition; bead-based diffusometry.” (Page1, Line 32-33)
“500-nm of PS-modified particles were conjugated with 1 µm of Janus particles to form an immunocomplex in the presence of the target antigen, thereby increasing the particle diameter.” (Page 3, Line 99-101)
“Subsequently, particle displacement was used to estimate the translational diffusivity within a predefined time interval (∆t).” (Page 3, Line 120-121)
“1-μm fluorescent PS particles (F13083, ThermoFisher) were first mixed with 95% ethanol and then spread over a glass slide by drop deposition to form a single layer of particles and obtain the desired Janus particles.” (Page 6, Line 193-195)
“These values were close to the theoretical value of 50% and 16%, which referred to 2 and 6 mPa.s” (Page 13, Line 413-414)
Comment 5:
I think that figure S1 from supplementary is a nice summary of the method and should appear in the main text where it will help the reader follow authors’ ideas.
Response 5:
Upon the reviewer’s suggestion, Figure S1 and the relevant caption were moved to the main text. Relevant change was also accomplished accordingly. (Page 5, Line 163-170)
Comment 6:
Line 92: How difficult is it to functionalize other antigens? A brief comment on that would be useful to reader.
Response 6:
The target antigen can be changed by replacing the conjugated antibodies on both the Janus particles and the PS nanoparticles. Conjugating Janus particles and PS nanoparticles with antibodies can be simply achieved by following standard protocols provided by their suppliers. The assays have been addressed in the main text. A brief statement regarding the change of the target antigen was added in the manuscript.
“The gold coating of the Janus particles partially blocked the fluorescence and enabled antibody conjugation via the thiol bond. Accordingly, monoclonal anti-IFN-γ IgG (orb26592, Biorbyt) could be conjugated with the Janus particles through a gold conjugation kit (ab154873, Abcam). The conjugate reaction took place in a 10-mM amine-free buffer, with a pH range between 6.5 and 8.5 for all antibodies. The antibodies were diluted to 0.1 mg/mL of the diluent buffer in the kit. Twelve microliters of the diluent antibodies were mixed with 42 µL of the gold reaction buffer. Subsequently, the mixture was incubated with Janus particles in a shaker (400 rpm) for 20 min at room temperature (~25 °C). Five microliters of the gold quencher buffer were mixed gently after incubation. The antibodies were covalently attached on the surface of Janus particles through lysine residues.
In amplifying particle geometric changes, 500-nm PS particles (07763-5, Polysciences) were further used to form a sandwiched immunocomplex in the presence of target antigens. Consequently, monoclonal anti-IFN-γ IgG was conjugated with the amine-modified 1 µm PS particles. Ten microliters of the particle suspension were washed with 200 µL of 50 mM 2-(N-morpholino) ethanesulfonic acid buffer (MES buffer) at pH 5.5. Meanwhile, 10 µL of 0.5 mg/mL antibodies was activated by mixing 2 µL of 10 mg/mL 1-ethyl-3-(3-dimethylaminopropyl)-carbodiimide, 4 µL of 10 mg/mL N-hydroxysuccinimide, and 30 µL of MES buffer in a shaker for 15 min at room temperature. Next, the mixture was incubated with particle suspension in a shaker (800 rpm) for 4 h at 4 °C. After storing at 4 °C overnight, the suspension was washed with PBS with 1% Tween-20 (PBST) three times and then blocked with 1% BSA for 1 h.
After preparing the functionalized Janus particles and the modified 500-nm PS particles, 20 µL of the functionalized Janus particle suspension was incubated with 10 mL of different concentrations (1 pg/mL, 100 pg/mL, 10 ng/mL, and 1 µg/mL) of IFN-γ antigen (orb82062, Biorbyt) in a shaker for 1 h at room temperature. Subsequently, the mixture was washed three times with PBST to remove unbound antigens, followed by incubating with 10 µL of the modified PS particle suspension for 1 h at room temperature. Afterward, the Janus particles were conjugated with the modified PS particles to form a sandwiched immunocomplex. The sandwiched immunocomplexed particle diameter basically grew with the increase of antigen concentration (Supporting Information 2)” (Page 6, Line 207-237)
“The target antigen can be changed by replacing the conjugated antibodies on both the Janus particles and the PS particles.” (Page 6, Line 237-238)
Comment 7:
Line 122: Can you define ‘circularity’?
Response 7:
The “circularity” of each particle image was defined as the squared circumference perimeter of a pattern divided by 4 times π times the area of the pattern. Circularity that approaches 1 means the shape is close to a perfect circle. The definition of circularity was added in the manuscript to clarify this point.
“The circularity of each particle image was defined as follows:
(4)
The circularity that approaches 1 means the pattern is close to a perfect circle.” (Page 4, Line 127-129)
Comment 8:
Line 124: ‘overlaps from adjacent particles were excluded from the calculation when their distances between two particles were shorter than 100 pixels’.
I am not sure to understand, how were they excluded?
Response:
We apologize for the unclear statement. Actually, before proceeding with particle tracking, all particles’ coordinates were located in advance. Only those particles which have no neighbor particles within a radius of 100 pixels were selected to be tracked for their displacements and blinking frequencies. This action can be simply achieved by running the function “pdist” built in MATLAB. The supporting interpretation was added in the main text.
“In addition, overlaps from adjacent particles were excluded from the calculation when their distances between two particles were shorter than 100 pixels. To this end, all particles’ coordinates were located in advance. Subsequently, distances between any two particles were obtained. This action was simply achieved by running the function “pdist” built in MATLAB. ” (Page 4, Line 129-134)
Comment 9:
Line 222: ‘and their relationship was approximated by a four-parameter exponential’
Is this approximation an empirical fitting or analytically derived?
Response:
This approximation is based on the empirical fitting. According to the reference1, the formula is expressed as:
where A, B, C, and D are constants of water, which are experimentally determined to be1.8610-11 mPa.s, 4209 K, 0.04527 K-1, and -3.37610-5 K-2, respectively. Notably, A, B, C, and D are constants subjected to different solutions. The additional information was added in the supporting information.
“This approximation is based on the empirical fitting. According to the reference1, the formula is expressed as:
where A, B, C, and D are constants of water, which are experimentally determined to be1.8610-11 mPa.s, 4209 K, 0.04527 K-1, and -3.37610-5 K-2, respectively. Notably, A, B, C, and D are constants subjected to different solutions.” (Supporting Information 3)
Comment 10: Line 227: the equation is not correct. $\langle x \rangle$ stands for the average of x that is 0. The quantity that is calculated is the standard deviation of displacement $\sqrt{\langle x^2\rangle}$.
Response:
We thank the reviewer for his/her reminder. The equation was corrected as follows.
” (Page 7, Line 253)
Comment 11:
Line 253: Is the detection of the best IMF manual or automatic?
Response:
The detection of the best IMF would work automatically through the cross-correlation algorithm. A sentence was added in the manuscript to point out this information.
“The similarity values were automatically calculated and compared on Matlab.” (Page 7, Line 274-275)
Comment 12:
Video S2: does not work on my VLC
Response:
The video format has been changed. Now it’s uncompressed and should be able to work properly for all platforms.
Comment 13:
Minor English :
“and applied in a wide variety of applications”-> Repetition of ‘apply’
“Consequently, the rotational and translational diffusivities were delicately correlated to remove the unfavorable background noises.” -> Correlated does not work here.
Response:
We thank the reviewer for his/her kind reminders, The inappropriate sentences were corrected as follows:
“The former is commonly investigated in a wide variety of applications” (Page 1, Line 37-38)
“Subsequently, the unfavorable background noises were removed by dividing the rotational diffusivity by the translational diffusivity.” (Page 15, Line 475-476)
Reference:
- S. Chuang, Y.J. Chen, H.P. Cheng, Enhanced diffusometric immunosensing with grafted gold nanoparticles for detection of diabetic retinopathy biomarker tumor necrosis factor-alpha, Biosensors & Bioelectronics. 101 (2018) 75–83. https://doi.org/10.1016/j.bios.2017.10.002.
- C. Wang, S.W. Chi, T.H. Yang, H.S. Chuang, Label-Free Monitoring of Microorganisms and Their Responses to Antibiotics Based on Self-Powered Microbead Sensors, Acs Sensors. 3 (2018) 2182–2190. https://doi.org/10.1021/acssensors.8b00790.
- P. Cheng, H.S. Chuang, Rapid and Sensitive Nano-Immunosensors for Botulinum, Acs Sensors. 4 (2019) 1754–1760. https://doi.org/10.1021/acssensors.9b00644.
- C. Wang, S.W. Chi, D.B. Shieh, H.S. Chuang, Development of a self-driving bioassay based on diffusion for simple detection of microorganisms, Sensors and Actuators B-Chemical. 278 (2019) 140–146. https://doi.org/10.1016/j.snb.2018.09.087.
- C. Wang, Y.C. Tung, K. Ichiki, H. Sakamoto, T.H. Yang, S. Suye, H.S. Chuang, Culture-free detection of methicillin-resistant Staphylococcus aureus by using self-driving diffusometric DNA nanosensors, Biosensors & Bioelectronics. 148 (2020). https://doi.org/ARTN 111817 10.1016/j.bios.2019.111817.
- -L. Chen, H.-S. Chuang, Trace Biomolecule Detection with Functionalized Janus Particles by Rotational Diffusion, Analytical Chemistry. 92 (2020). https://doi.org/10.1021/acs.analchem.0c01733.
- N. Clayton, G.D. Berglund, J.C. Linnes, T.L. Kinzer-Ursem, S.T. Wereley, DNA Microviscosity Characterization with Particle Diffusometry for Downstream DNA Detection Applications, Analytical Chemistry. 89 (2017) 13334–13341. https://doi.org/10.1021/acs.analchem.7b03513.
- Robert C.Reid; John M. rausnitz; Poling, Bruce E. (1987), The Properties of Gases and Liquids, McGraw-Hill Book Company, ISBN0-07-051799-1

Reviewer 2 Report
The authors study a simple self-compensated algorithm based on delicate removal of temperature and fluid viscosity variations through particle tracking. Consequently, translational Brownian motion was expressed with regard to particle trajectory, whereas rotational Brownian motion was expressed with regard to blinking signal from Janus particles. The algorithm was verified simulatively and experimentally in temperature (10 °C to 40 °C) and viscosity-controlled (1 mPa.s to 5 mPa.s) fields. In an evaluation of biosensing for a target protein, IFN- , the limit of detection of self-compensated diffusometry reached 0.45 pg/mL, whereas its uncertainty of viscosity and temperature were 96 and 15-fold lower than pure rotational Brownian motion counterpart, respectively. Result indicated the low-uncertainty and high-accuracy biosensing capability resulting from the self-compensated algorithm. This research will provide a potential alternative to future similar bead-based immunosensing, which requires ultra-high stability and sensitivity. The numerical results are quite impressive. So based on these comments, I recommend this paper for publication.
Author Response
Editorial Board
Biosensors
May. 21, 2022
Dear Editor,
On behalf of the authors, I would like to submit our revised manuscript entitled “Development of a self viscosity and temperature compensated technique for highly stable and highly sensitive bead-based diffusometry” (ID: biosensors-1709634) in response to the reviewers’ comments. As requested by the reviewers, we have clarified all the minor concerns. Our point-by-point responses to the reviewers are also attached below and all the changes are highlighted throughout the annotated manuscript. We hope, with the improvement, you will now find our manuscript suitable for publication in Biosensors. Finally, we thank you and the reviewers for your precious time and advice. Please feel free to contact us anytime with any questions.
Sincerely,
Corresponding Author Information:
Han-Sheng Chuang, Professor
Tel:+886 6-2757575#63433, Fax:+886 6-2343270, oswaldchuang@mail.ncku.edu.tw
Department of Biomedical Engineering, National Cheng Kung University, Tainan, Taiwan
Point by point responses to reviewers’ comments
Reviewer #2
General Comment:
The authors study a simple self-compensated algorithm based on delicate removal of temperature and fluid viscosity variations through particle tracking. Consequently, translational Brownian motion was expressed with regard to particle trajectory, whereas rotational Brownian motion was expressed with regard to blinking signal from Janus particles. The algorithm was verified simulatively and experimentally in temperature (10 °C to 40 °C) and viscosity-controlled (1 mPa.s to 5 mPa.s) fields. In an evaluation of biosensing for a target protein, IFN- , the limit of detection of self-compensated diffusometry reached 0.45 pg/mL, whereas its uncertainty of viscosity and temperature were 96 and 15-fold lower than pure rotational Brownian motion counterpart, respectively. Result indicated the low-uncertainty and high-accuracy biosensing capability resulting from the self-compensated algorithm. This research will provide a potential alternative to future similar bead-based immunosensing, which requires ultra-high stability and sensitivity. The numerical results are quite impressive. So based on these comments, I recommend this paper for publication.
Response:
We truly appreciate the reviewer’s positive feedback.

Reviewer 3 Report
This is potentially good work, however the title and abstract are misleading, at least for me. The paper is mainly about experimental research. The title contains words about algorithms, but inside the article there is no special section about algorithm. The description is contained in several sections, and main details are placed in the "Result and discussion". I couldn't figure out what the authors meant by "simulated," where it was simulated, what tools were used to simulate it, or what methods and parameters of methods were used. For me, simulation is always connected with computation. Maybe authors are used to other terminology, but it would be better to include the definition of "simulation" in the paper.
What compensates the algorithm and why it is self-compensated also is remained unclear to me. It should be described more explicitly.
It is not clearly explained why the author adopted the algorithm from the paper. For example, at the first glance, FFT can be used to extract dominated frequency directly without using the empirical mode decomposition. The explanation of author’s reasons for choosing one method or another has to be added to the paper.
In Figure 2, it would be preferable if the authors displayed an IMF with a high similarity value rather than one IMF with an average similarity.
Minor remarks
There are some misprints. I noticed "Stoke" instead of "Strokes" (and Strokes-Einstein, Stokes–Einstein Debye). In several parts of the text, Greek letters are disappeared. For example, please check units in this phrase: "made of 1 m fluorescent core polystyrene". Also, "(IFN- . Five hundred nanometers of PS-modified 93 particles was conjugated with 1 m of J" and so on. By the way, it should be "particles were" in the previous sentence.
It seems to me that articles (a/the) are omitted frequently without grammatical reason, but I am not a native English speaker and can only recommend carefully checking this aspect of the text. For example, "of the rotational Brownian motion?" instead of "of rotational Brownian motion" and so on.
Indentation after formulas is usually absent, however authors added an indent. Please, check the rules of the journal regarding this point.
Please, add figures from which the reader can understand a sense in the paragraph "2.1.1 Particle Tracking for Trajectory" without watching supporting video.
Author Response
Editorial Board
Biosensors
May. 21, 2022
Dear Editor,
On behalf of the authors, I would like to submit our revised manuscript entitled “Development of a self viscosity and temperature compensated technique for highly stable and highly sensitive bead-based diffusometry” (ID: biosensors-1709634) in response to the reviewers’ comments. As requested by the reviewers, we have clarified all the minor concerns. Our point-by-point responses to the reviewers are also attached below and all the changes are highlighted throughout the annotated manuscript. We hope, with the improvement, you will now find our manuscript suitable for publication in Biosensors. Finally, we thank you and the reviewers for your precious time and advice. Please feel free to contact us anytime with any questions.
Sincerely,
Corresponding Author Information:
Han-Sheng Chuang, Professor
Tel:+886 6-2757575#63433, Fax:+886 6-2343270, oswaldchuang@mail.ncku.edu.tw
Department of Biomedical Engineering, National Cheng Kung University, Tainan, Taiwan
Point by point responses to reviewers’ comments
Reviewer #3
General Comment:
This is potentially good work, however the title and abstract are misleading, at least for me. The paper is mainly about experimental research. The title contains words about algorithms, but inside the article there is no special section about algorithm. The description is contained in several sections, and main details are placed in the "Result and discussion". I couldn't figure out what the authors meant by "simulated," where it was simulated, what tools were used to simulate it, or what methods and parameters of methods were used. For me, simulation is always connected with computation. Maybe authors are used to other terminology, but it would be better to include the definition of "simulation" in the paper.
Response:
We truly appreciate the reviewer’s positive feedback. Regarding some unclear aspects of the manuscript, we have revised them carefully in response to the reviewer’s comments point by point. In addition, we changed the “self-compensated algorithm” to “self viscosity and temperature compensated technique” to better reflect the work done in this research. We simulated the rotational Brownian motion and the translational Brownian motion to verify this technique. Both the rotational Brownian motion and the translational Brownian motion were simulated by a self-developed MATLAB program. To clarify the reviewer’s concern, the title of section 3.2 was changed to “Evaluations of Simulated Bead-based Brownian Motion with Viscosity and Temperature Changes”.
Comment 1:
What compensates the algorithm and why it is self-compensated also is remained unclear to me. It should be described more explicitly.
Response 1:
According to the past literature, 1-7 Brownian motion could be used to detect proteins, DNAs, or microorganisms, which show lower LODs than their counterparts, such as ELISA or western blot. However, Brownian motion, if not well controlled, can be easily altered by the background noises, such as fluid viscosity and temperature fluctuations. Therefore, we proposed a self temperature and viscosity compensated technique to deal with the problem. The proposed technique requires to simultaneously measure both the translational and rotational Brownian motions from the same particle. Based on the Stokes-Einstein equation and Stokes-Einstein Debye equation, their background viscosity and temperature can be eliminated by simply dividing one by another. Accordingly, an equivalent particle diameter was eventually derived. The “self” denotes both translational and rotational diffusivity signals extracted from the same particle, while the “compensated” indicates the background noise was eliminated through those signals. An explicit explanation was added in the manuscript as requested.
“In addressing the problem, we proposed a method, namely, self viscosity and temperature compensated technique, which combined translational and rotational Brownian motion in this research. Based on the Stokes-Einstein equation and Stokes-Einstein Debye equation, their background viscosity and temperature can be eliminated by dividing the translational diffusivity by the rotational diffusivity. The ratio of the translational diffusivity to the rotational diffusivity will be proportional to the squared particle diameter. The proposed technique was verified on the basis of experimental and simulated Brownian motion.” (Page 2, Line 61-68)
Comment 2:
It is not clearly explained why the author adopted the algorithm from the paper. For example, at the first glance, FFT can be used to extract dominated frequency directly without using the empirical mode decomposition. The explanation of author’s reasons for choosing one method or another has to be added to the paper.
Response:
Indeed, we used FFT to process the signal at first. There exists an optimal recipe after a couple of trials and errors in the study. However, the results showed a wide distribution of frequency dominated. The low-frequency blinking signal was mixed with high-frequency noise such as 60 Hz of electric noise, 50 Hz of capture noise, or white noise. Moreover, when the particles were suspended in high viscosity solution, the in-focus and out-focus intensity changed, and the blinking signal became ambiguous. The blinking frequency would not be able to be distinguished through FFT especially if the particles were suspended in high viscosity solution. A well-designed filter is necessary for this circumstance. To reduce most noise of blinking signal, we finally found EMD is the other way to decompose the signal into several IMFs from high frequency to low frequency. To precisely extract the blinking signal, each of the IMFs was compared with the original blinking signal. Therefore, the IMF showing the highest similarity indicated the optimal waveform that carried the pure blinking signal. The following sentences were added in the paper.
“Notably, low-frequency noise was not able to be distinguished through Fast Fourier Transform (FFT) especially if the particles were suspended in high viscosity solution. A well-designed filter is necessary for this circumstance.” (Page 4, Line 148-150)
Comment 3:
In Figure 2, it would be preferable if the authors displayed an IMF with a high similarity value rather than one IMF with an average similarity.
Response:
We thank the reviewer for the suggestion. The similarity of IMF in Figure 3 showed the similarity value of one IMF from the left side of Figure 3 than average similarity. Indeed, we used fixed (average similarity of IMF) IMF to analyze the data at the beginning. However, the viscosity changes according to ambient temperature and the edge of the glass slide. Especially since the particles were simulated under higher viscosity, the signal became ambiguous under IMF 3 and IMF 4. If we analyzed IMF 4 in each particle, which gives us the wrong information about the blinking signal. Therefore, we used a cross-correlation algorithm to compare each IMF to its own original blinking signal. The correct frequency of each particle could be estimated through this comparison. On the other hand, Figure 3 could be a comparison with Figure 6 to show the simulation result is comparable with the experimental result. To clarify the concerning the caption of Figure 3 changed as follow:
“Figure 3. EMD decomposes the noise blinking signal. (a) The signal without noise under 1 mPa.s and 25 °C. Three of IMFs are shown out of 4 IMFs. (b) The similarity of each IMF to the original signal without noise (c) The signal with noise under 1 mPa.s and 25 °C. Three of IMFs are shown out of 6 IMFs. (d) The similarity of each IMF to the original signal with noise. (e) The signal with noise under 5 mPa.s of viscosity. Four of IMFs are shown out of 5 IMFs. (f) The similarity of each IMF to the original signal w/ noise in the 5 mPa.s group.” (Page 8, Line 294-301)
Comment 4:
There are some misprints. I noticed "Stoke" instead of "Strokes" (and Strokes-Einstein, Stokes-Einstein Debye). In several parts of the text, Greek letters are disappeared. For example, please check units in this phrase: "made of 1 m fluorescent core polystyrene". Also, "(IFN- . Five hundred nanometers of PS-modified 93 particles was conjugated with 1 m of J" and so on. By the way, it should be "particles were" in the previous sentence.
Response:
We deeply apologize for the incorrect unit and symbols in the text. The incorrect symbols and units, such as γ, µ, Stokes, were corrected in the manuscript.
“Keywords: Rotational Brownian motion; translational Brownian motion; particle tracking; empirical mode decomposition; bead-based diffusometry.” (Page1, Line 32-33)
“500-nm of PS-modified particles were conjugated with 1 µm of Janus particles to form an immunocomplex in the presence of the target antigen, thereby increasing the particle diameter.” (Page 3, Line 99-101)
“Subsequently, particle displacement was used to estimate the translational diffusivity within a predefined time interval (∆t).” (Page 3, Line 120-121)
“1-μm fluorescent PS particles (F13083, ThermoFisher) were first mixed with 95% ethanol and then spread over a glass slide by drop deposition to form a single layer of particles and obtain the desired Janus particles.” (Page 6, Line 193-195)
“These values were close to the theoretical value of 50% and 16%, which referred to 2 and 6 mPa.s” (Page 13, Line 413-414)
Comment 5:
It seems to me that articles (a/the) are omitted frequently without grammatical reason, but I am not a native English speaker and can only recommend carefully checking this aspect of the text. For example, "of the rotational Brownian motion?" instead of "of rotational Brownian motion" and so on.
Response:
We thank the reviewer for his/her kind reminder. All the errors, to the best of our knowledge, were corrected in the main text. Some of them are listed as follows:
“However, the rotational Brownian motion is inversely proportional to cubic particle diameter” (Page 1, Line 39-40)
“This ultra-high sensitivity promotes the capability of the rotational Brownian motion to deal with trace target detection.” (Page 1, Line 40-42)
“Alternatively, colloidal dimers, tetrahedral clusters, diamond nanoparticles, and spherical Janus particles are extensively developed to facilitate the investigation of the rotational Brownian motion.” (Page 2, Line 48-50)
“Considering that the rotational Brownian motion based on Janus particles yields signal only associated with temporal intensity variation, the requirement of image quality can be simplified, thereby lowering the uncertainty of angular change.” (Page 2, Line 53-55)
Comment 6:
Indentation after formulas is usually absent, however authors added an indent. Please, check the rules of the journal regarding this point.
Response:
We are sorry for the inconsistent format. All the formula formats were modified in the following pages and lines. (Page 3, Line 108; Page 3, Line 111-112; Page 7, Line 246)
Comment 7:
Please, add figures from which the reader can understand a sense in the paragraph "2.1.1 Particle Tracking for Trajectory" without watching supporting video.
Response:
Upon the reviewer’s suggestion, Figure S1 and the relevant caption were added to the main text. The scheme in Figure S1 was also remodified to reflect the procedure. (Page 5, Line 159-166)
“The translational diffusivity was obtained by tracking particles for their trajectories. Particle displacement was derived from their coordinates in two different time points multiplied by the pixel size (Figure 2). Subsequently, particle displacement was used to estimate the translational diffusivity within a predefined time interval (∆t). Five hundred consecutive images were recorded in 10 s with a frame rate of 50 Hz.
In the particle tracking, all images were binarized (black: 0; white: 1) to facilitate the search for particle outlines. The coordinate of each particle was defined in the centroid of the particle image area. Notably, the white areas smaller than 10 pixels and their circularity lower than 0.35 (maximum value of circularity: 1) were excluded to avoid some background noises.” (Page 3, Line 118-127)
“
Figure 2. The schematic of the self viscosity and temperature compensated technique step. The particle tracking was used to evaluate the translational and the rotational diffusivities. The particle images were binarized by thresholding to define circularity and of each particle. The coordinates of each particle were defined after removing the close particles. The blinking signal of the particle was captured according to particle coordinates. To eliminate the noise of the blinking signal, EMD was applied to decompose the blinking signal into several IMFs. The particle diameter was derived by dividing the translational diffusivity by the rotational diffusivity.” (Figure 2) (Page 5, Line163-170)
Reference:
- S. Chuang, Y.J. Chen, H.P. Cheng, Enhanced diffusometric immunosensing with grafted gold nanoparticles for detection of diabetic retinopathy biomarker tumor necrosis factor-alpha, Biosensors & Bioelectronics. 101 (2018) 75–83. https://doi.org/10.1016/j.bios.2017.10.002.
- C. Wang, S.W. Chi, T.H. Yang, H.S. Chuang, Label-Free Monitoring of Microorganisms and Their Responses to Antibiotics Based on Self-Powered Microbead Sensors, Acs Sensors. 3 (2018) 2182–2190. https://doi.org/10.1021/acssensors.8b00790.
- P. Cheng, H.S. Chuang, Rapid and Sensitive Nano-Immunosensors for Botulinum, Acs Sensors. 4 (2019) 1754–1760. https://doi.org/10.1021/acssensors.9b00644.
- C. Wang, S.W. Chi, D.B. Shieh, H.S. Chuang, Development of a self-driving bioassay based on diffusion for simple detection of microorganisms, Sensors and Actuators B-Chemical. 278 (2019) 140–146. https://doi.org/10.1016/j.snb.2018.09.087.
- C. Wang, Y.C. Tung, K. Ichiki, H. Sakamoto, T.H. Yang, S. Suye, H.S. Chuang, Culture-free detection of methicillin-resistant Staphylococcus aureus by using self-driving diffusometric DNA nanosensors, Biosensors & Bioelectronics. 148 (2020). https://doi.org/ARTN 111817 10.1016/j.bios.2019.111817.
- -L. Chen, H.-S. Chuang, Trace Biomolecule Detection with Functionalized Janus Particles by Rotational Diffusion, Analytical Chemistry. 92 (2020). https://doi.org/10.1021/acs.analchem.0c01733.
- N. Clayton, G.D. Berglund, J.C. Linnes, T.L. Kinzer-Ursem, S.T. Wereley, DNA Microviscosity Characterization with Particle Diffusometry for Downstream DNA Detection Applications, Analytical Chemistry. 89 (2017) 13334–13341. https://doi.org/10.1021/acs.analchem.7b03513.
